# Combined Bipolar Radiofrequency and Non-Crosslinked Hyaluronic Acid Mesotherapy Protocol to Improve Skin Appearance and Epidermal Barrier Function: A Pilot Study

**DOI:** 10.3390/ph16081145

**Published:** 2023-08-12

**Authors:** Anna Płatkowska, Szymon Korzekwa, Bartłomiej Łukasik, Nicola Zerbinati

**Affiliations:** 1Anclara Health & Aesthetics, 02624 Warsaw, Poland; skorzekwa@anclara.pl; 2Medical Department, Matex Lab Switzerland SA, 1228 Geneva, Switzerland; bartlomiej.lukasik@neauvia.com; 3Dermatologic Unit, University of Insubria, 21100 Varese, Italy; nicola.zerbinati@uninsubria.it

**Keywords:** skin aging, bipolar radiofrequency, non-crosslinked hyaluronic acid (HA) mesotherapy

## Abstract

Background: Age-associated changes in epidermal hydration, pigmentation, thickness and cell renewal influence skin appearance and can lead to laxity, dryness and poor skin tone. The aim of this pilot study was to assess the synergistic effects of a new bipolar radiofrequency plus non-crosslinked hyaluronic acid (HA) mesotherapy protocol compared with radiofrequency alone on skin appearance and markers of epidermal function. Methods: This prospective, single-center, split-face pilot study recruited women aged 25–65 years with dryness and laxity of the facial skin defined by a trans-epidermal water loss (TEWL) value of ≥26 g/m^2^/h. Subjects were treated with a bipolar radiofrequency device on both sides of the face. This was immediately followed by needle hyaluronic acid (HA) treatment on one side of the face with 2.5 mL of a non-crosslinked HA. Photographic documentation, analysis of epidermal barrier function parameters, and high frequency (HF) ultrasound analysis were performed prior to treatment and at 28 days. Results: Twenty female subjects with a mean age of 46 (range 29 to 54) years and dry and lax facial skin were included. TEWL was reduced and skin hydration improved to a greater extent with the combined radiofrequency plus mesotherapy protocol compared with radiofrequency alone (−5.8% vs. +3.9% and +23.1% vs. +1.0%, respectively). The combined protocol was also associated with greater improvements in melanin (−7.5% vs. −1.5%) and erythema values (−7.2% vs. +3.0%), respectively. Ultrasound measures of epidermal thickness and epidermal density were greater after the combined protocol compared with radiofrequency alone (12.0% vs. 5.6% and 57.7% vs. 7.1%, respectively). Both treatments were well-tolerated. Conclusions: The combined bipolar radiofrequency and HA mesotherapy protocol provided greater improvements in skin hydration, firmness and tone compared with radiofrequency alone. The combination treatment was also associated with greater epidermal thickness and density and increased keratinocyte differentiation suggesting a synergistic effect of both treatments on epidermal homeostasis and barrier function. Both treatments were well-tolerated and led to improvements in facial appearance.

## 1. Introduction

Skin aging is one of the most visible indicators of advancing age with the first changes to the face generally becoming apparent in the third decade of life [1]. As a result, there is high consumer demand for anti-ageing treatments in all age groups. Individuals are increasingly requesting aesthetic procedures that improve skin quality as part of a global facial approach that addresses skin laxity, hydration and overall skin tone, as well as reducing lines and wrinkles.

The aging of human skin is a complex process that involves genetically-determined intrinsic factors and age-related hormonal changes, the effects of which are accelerated by cumulative exposure to external environmental factors, mainly ultraviolet (UV) light exposure, but also factors such as air pollution, smoking, and poor nutrition [2]. 

The skin comprises the dermis and epidermis overlying an inner layer of subcutaneous tissue. The dermis consists mostly of extracellular matrix (ECM) with collagen, elastin and glycosaminoglycans as the principal components, all of which are produced by fibroblasts. The dermal ECM is arranged in a dense network of interlinked collagen (types I and III) and elastin fibers containing water-retaining proteoglycans, and provides the skin with both strong mechanical resistance and elasticity [2]. The outer layer of skin, the epidermis, is attached to the dermis by specialized proteins of the basement membrane. It is a stratified epithelium consisting mainly of keratinocytes, which undergo terminal differentiation and form the stratum corneum, which serves as a barrier against environmental damage. 

Intrinsic skin aging is characterized by a progressive loss of tissue with a thinning of all skin layers [2,3]. In the dermis, the collagen network undergoes re-organization, resulting in a reduction in collagen, proteoglycan and hyaluronic acid (HA) content, accompanied by an increase in denatured and degraded collagen. These changes lead to a loss of ECM volume, and reduced mechanical tension on the fibroblasts, which is required for optimal regulation of collagen and other ECM components. The dermis is particularly affected by ageing as it has a low turn-over compared to other skin layers [4]. The most noticeable microstructural modifications of the dermis with age include decreased collagen and elastin content, a reduction in type I collagen leading to an increased ratio of type III to type I, and a deterioration of ECM proteoglycans.

Within the epidermis, the turnover of keratinocytes is reduced resulting in decreased epidermal thickness, flattening of the dermal-epidermal junction, and a reduced capacity to rapidly restore the skin barrier [2,3].

Extrinsic aging further impairs the structural integrity of the skin creating disorganized and fragmented collagen networks [5], and increased amounts of abnormal and disordered elastin [6]. Skin also requires adequate levels of hydration and transepidermal water loss (TEWL) must be carefully regulated. The prevention of TEWL is dependent on two major components: the presence of natural hygroscopic agents, such as HA, and an effective stratum corneum epidermal barrier. The relative permeability of the different epithelia reflect the extent of their exposure to the external environment with skin epithelia being the least permeable and intestinal the most [7]. A greater susceptibility to dry, rough and fragile skin in older individuals is associated with degradation and slower recovery of skin epithelial barrier function. 

Hyaluronic acid is abundant in the dermis, where it contributes to the turgor and flexibility of healthy skin. It is also found in the epidermis where in addition to acting as a humectant, it is involved in maintaining normal stratum corneum structure and epidermal barrier function and serves a number of bioregulatory functions [8,9]. Both intrinsic and extrinsic factors appear to downregulate HA synthases, resulting in a reduction in HA levels, particularly in the epidermis [10,11,12].

The epidermis is important from an aesthetic standpoint, because it is the layer that provides the skin with its texture and moisture, and contributes to skin color. If the surface of the epidermis is dry or rough, the skin appears aged. A healthy epidermis continually regenerates itself to maintain a functional barrier and to respond to cutaneous insults. It is primarily composed of keratinocytes, which migrate upwards from basal to cornified layers and undergo a program of differentiation that ultimately leads to their transformation into corneocytes before they desquamate from the skin surface [13]. With age, epidermal cell turnover decreases and there is reduced nutrient transfer between the dermis and epidermis, which contributes to less effective desquamation and increased skin fragility [14]. 

Each stage of keratinocyte differentiation is associated with the expression of specific proteins: Ki67 (marker of cell proliferation), K10 (required for keratin filaments), and K17 (regulates many biological processes, including cell proliferation and growth, skin inflammation and hair follicle cycling) [15]. Epidermal homeostasis is dependent on the regenerative capacity of these cells and the complex signaling pathways that orchestrate it. Aged epidermis displays multiple alterations in keratinocyte differentiation and function which have been linked to a decrease in the integrity of the stratum corneum and a delayed ability to recover in older compared with younger age groups [16]. Reduced levels of hyaluronic acid are also observed in aged skin [17], and studies have shown that topical application of hyaluronic acid can stimulate keratinocyte differentiation as well as lipid production, leading to enhanced epidermal permeability barrier function [18]. 

To counteract these aging-related skin changes, a number of strategies have been developed that aim to rejuvenate the dermal and epidermal layers to reflect those of young and healthy skin [19]. Topical agents can help improve skin barrier function, but to stop the degradation and rejuvenate the skin primary structural constituents, such as collagen and elastin, technologies that deliver treatments deeper into the skin are required. These include chemical peelings, visible light devices, intense pulsed light, ablative and nonablative laser photo–rejuvenation, radiofrequency, and injectable skin biostimulation and rejuvenation in the form of mesotherapy. 

Radiofrequency and mesotherapy protocols, in particular, provide skin rejuvenating effects with minimal to no downtime. Radiofrequency produces heat when the electrical resistance of tissues converts the electric current to thermal energy. In vitro studies have shown that radiofrequency currents can trigger the proliferation and migration of keratinocytes and fibroblasts, the main cell types involved in skin regeneration [20]. Both monopolar and bipolar radiofrequency treatments have demonstrated direct collagen contraction and immediate skin tightening. Subsequent remodeling and reorientation of collagen bundles, and the formation of new collagen, is achieved over months of treatment, which leads to an improvement in skin laxity [21,22,23,24].

The introduction of HA-based formulations as multiple micro-injections promotes skin hydration as well as keratinocyte differentiation to provide an optimal environment for healthy skin homeostasis and protect against extrinsic aging-related risk factors [25,26,27,28].

Combining treatments that act on multiple aspects of skin aging allows physicians to address the visible signs based on an understanding of their underlying cause, and is becoming increasingly popular to maximize outcomes and improve patient satisfaction. The main objective of this pilot study was to assess the synergistic effects of a new bipolar radiofrequency plus non-crosslinked hyaluronic acid mesotherapy protocol compared with radiofrequency alone on skin appearance and markers of epidermal function.

## 2. Results

The study recruited 20 female subjects, mean age 46 years (range 29 to 54 years), with Fitzpatrick skin phototype I–IV. All women presented with dry and lax facial skin with a TEWL value of at least 26 g/m^2^/h. 

All subjects completed the study, and showed clear clinical improvement of skin tightening and rhytides in the cheek, chin, crow’s feet and forehead regions (Figure 1).

TEWL was reduced and skin hydration improved to a greater extent with the combined radiofrequency plus mesotherapy protocol compared with radiofrequency alone (−5.8% vs. +3.9% and +23.1% vs. +1.0%, respectively) (Table 1).

### 2.1. Skin Pigmentation

Subjects’ melanin and erythema values, reflecting the subjects’ skin tone, were improved with the combined protocol compared with radiofrequency alone (Table 1). Further analysis of skin pigmentation assessed with the Smart Mirror-Skin Analysis System confirmed that the addition of mesotherapy to the radiofrequency protocol resulted in greater improvements in all parameters analyzed compared with radiofrequency alone (Table 1).

### 2.2. Epidermal Thickness and Density

Mean changes in skin thickness and epidermal thickness measured by ultrasound were approximately twofold greater after the combined protocol compared with radiofrequency alone (Table 1). 

The change in epidermal density was particularly pronounced with the combined protocol with an improvement of 57.7% compared with baseline, versus 7.1% compared with baseline for radiofrequency alone.

### 2.3. Adverse Effects

All patients tolerated the procedures well with no adverse events related to treatment. No palpable nodules, erythema, or post-inflammatory hyperpigmentation were observed. 

## 3. Materials and Methods

### 3.1. Participant Selection

This prospective, single-center, split-face pilot study recruited women aged 25–65 years in good general health with dryness and laxity of the facial skin defined by a TEWL value of at least 26 g/m^2^/h. Skin firmness was assessed by facial imaging and palpation, with the presence of gravitational wrinkles in the area of the tear trough, nasolabial folds, and marionette lines taken as evidence of a decrease in firmness. 

Participants with the following conditions or treatments in their medical history were excluded: current skin diseases (autoimmune disorders involving the skin, infections and inflammation); severe allergies; use of treatments that may influence skin healing in the 6 months prior to study entry (tetracyclines, immunosuppressants, systemic steroids, anticoagulants, cosmetics containing thyme extract, herbs containing St. John’s wort); facial aesthetic treatments involving fillers and/or energy-based devices in the 6 months prior to study entry; hypersensitivity to hyaluronic acid; and pregnancy or breastfeeding. Participants with a history of permanent metal implant devices or pacemakers were also excluded due to the use of radiofrequency.

### 3.2. Treatment Protocol

The study consisted of two visits, 4 weeks apart. Subjects were asked not to use moisturizing cosmetics on the day before each visit. Prior to performing facial imaging and skin measurements, subjects were asked to clean their face and remove all makeup. Subsequently, subjects were asked to sit quietly for 30 min to allow equilibration of their skin to the ambient conditions. At the first visit (V1), photographic documentation and analysis of the surface and color of the skin was performed. Objective assessment of epidermal barrier function was obtained using the Cutometer^®^ Dual MPA 580 (Courage + Khazaka electronic GmbH, Cologne, Germany) based on analysis of the following biophysical indicators: TEWL; skin hydration; melanin concentration; and intensity of erythema and skin elasticity. High frequency ultrasound analysis was also performed with images obtained at six points on both sides of the face covering the infraorbital, zygomatic, and corner of the mouth areas.

Participants were then treated with radiofrequency, according to manufacturer’s instructions, and adjusted to each patient. This was followed by needle mesotherapy treatment on one (randomly selected) side of the face with 2.5 mL of a non-cross-linked hyaluronic acid (Neauvia Hydro Deluxe, Matex Lab, Geneva, Switzerland).

At the second visit (V2), repeat photographic documentation, photographic analysis, analysis of skin biophysical indicators, and ultrasound analysis was performed, followed by needle mesotherapy on the side of the face not treated at V1.

### 3.3. Radiofrequency Treatment

Radiofrequency was performed once, prior to treatment with the mesotherapy product, using a Sectum device (Berger & Kraft Medical Sp. z o.o., Warsaw, Poland) and a FACE hand piece (Berger & Kraft Medical Sp. z o.o., Warsaw, Poland). Output frequency was 480 kHz. During the treatment, a current is released in a continuous mode and the applicator is set in motion mode using slow circular movement over the facial skin. Treatment times for the different areas of the face were as follows: left face cheek and chin (10–15 min), right face cheek and chin (10–15 min), forehead and crow’s feet (10 min). The energy was set to between 10 and 40 W and temperature to 38–41 °C and was adjusted based on patient feedback (comfortable or uncomfortable). The temperature for treatment of the forehead and crow’s feet was generally 1 °C lower than that used to treat the rest of the face.

### 3.4. Hyaluronic Acid Treatment

The non-crosslinked HA used in the study contained sodium hyaluronate 18 mg/mL, calcium hydroxyapatite microspheres (0.01%), and glycine and L-proline in buffered non-pyrogenic water. The product was supplied in a pre-filled 2.5 mL disposable syringe with a luer-lock connector. Mesotherapy treatment consisted of multiple intradermal or subcutaneous injections of HA using a 30G needle and the pre-filled syringe. To minimize pain, local anesthetic cream (EMLA^®^, Astra Zeneca, Cambridge, UK) was applied under occlusion for 30 min to the treated area before starting the procedure.

### 3.5. Post Procedure Recommendations

Subjects were advised not to apply pressure or use cosmetic products on the treated areas in the first 24 h after treatment. They were also advised to avoid surgery and dental procedures for 2 weeks after injection, and cosmetic procedures such as laser therapy or exfoliation of the treated area for 3 weeks after injection. They were recommended to avoid exposure to direct sunlight and UVA-B radiation (e.g., in a solarium) for 2 weeks after treatment.

### 3.6. Facial Skin Parameters

The facial skin of all patients was assessed before treatment and at 28 days using non-invasive cutometric, high-frequency ultrasound, and photographic analyses. Measurements of TEWL, skin hydration, melanin and erythema were obtained with Tewameter^®^, Corneometer^®^, and Mexameter^®^ probes, respectively (Courage and Khazaka Multi Probe Adaptor [MPA] System, Cologne, Germany). Changes in parameters compared with pretreatment values (baseline) are presented.

The Smart Mirror-Skin Analysis System (Langdai International Beauty Co., Ltd., Hong Kong, China), which uses the RGB, ultraviolet and pulsed light imaging spectrum, was used to assess a range of skin parameters including: RGB pore (larger than normal pore size indicating blocked pores), RGB spot (spot of darker than normal skin color), RGB wrinkle (current wrinkle condition), PL texture (current skin texture), UV porphyrin (sebum secretion and blackhead distribution), UV pigmentation (current dermal pigmentation), UV moisture (deep skin moisture state), red area, brown spot (skin metabolism condition), and UV damage (sun damage to deep skin layers).

Epidermal thickness (measured from the outermost epidermis [stratum corneum] to the basal cells) and epidermal density (resistance of medium to ultrasound) were assessed with a high frequency and high-resolution diagnostic ultrasound system (DUB^®^ SkinScanner, Taberna Pro Medicum, Lueneburg, Germany).

### 3.7. Ethical Considerations

The study was conducted in accordance with the Declaration of Helsinki and was approved by the Ethics Committee of Medical Chamber in Warsaw, Poland (protocol code: 05/22, 10 March 2022). Informed, written consent was obtained from all patients.

## 4. Discussion 

Aging skin is subject to a number of cellular and functional alterations that influence quality and appearance, and which are amenable to treatment. In this split-face pilot study, a treatment protocol combining bipolar radiofrequency with a non-crosslinked hyaluronic acid mesotherapy procedure improved epidermal barrier function parameters including stratum corneum hydration, TEWL, and pigmentation to a greater extent than treatment with radiofrequency alone. Ultrasound analysis confirmed that the combined protocol was also associated with significantly improved epidermal thickness and cell density compared with radiofrequency alone. 

TEWL is a key component when considering the skin barrier as it assesses stratum corneum function [29,30]. High TEWL values indicate that the permeability of the skin is increased, which means that its integrity is disrupted and skin hydration is reduced. Gradual dehydration of the epidermal layers starting with the stratum corneum can lead to the release of inflammatory mediators and disruption of epidermal differentiation, leading to further impairments in barrier function [31]. The combined treatment protocol in this study led to improved epidermal hydration and reduced TEWL. The hyaluronic acid mesotherapy component of the treatment replenished endogenous HA, levels of which are reduced with age and whose humectant properties are required to maintain epidermal water content. In animal studies, radiofrequency treatment has also been shown to increase epidermal HA, and to reverse age-related epidermal dysfunction [32]. It is hypothesized that when combined, the two treatments act synergistically to increase levels of epidermal HA and skin hydration.

The mechanical stress generated by the hyaluronic acid may also enhance the biochemical response of dermal cells. Mechanical tension is required by fibroblasts for optimal collagen and elastin homeostasis, but is reduced with age as HA content falls and skin collagen production becomes disordered and diminished. Skin biopsies obtained after injection of HA filler into human forearm indicate that mechanical stress generated by injection of the hyaluronic acid may enhance mechanical stretching and collagen production in dermal ECM [33].

With aging, there is a gradual decrease in the number of melanocytes, estimated at 10–20% per decade, which also become unevenly distributed in the epidermis [34,35]. The loss of functional melanocytes and alterations in the interactions between melanocytes and keratinocytes leads to protective barrier dysfunction against UV radiation [36]. Uneven skin tone as a result of melanin deposits and erythema was improved when mesotherapy was added to the radiofrequency protocol, but erythema was slightly increased with radiofrequency alone. The benefits of the combined treatment on skin appearance were confirmed by the Smart Mirror Skin-Analysis System, which showed that 28 days after initial treatment, a range of epidermal skin problems including large pore size, pigmentation spots and UV damage were improved compared with baseline. 

Aged human skin is characterized by a thinned epidermis, caused in part by decreased proliferation of basal keratinocytes [37]. Both epidermal thickness and overall skin thickness were improved with radiofrequency and HA mesotherapy. With the combined protocol, epidermal thickness was increased by 12% compared with baseline, twice that achieved with radiofrequency alone. In addition to improving skin laxity, epidermal thickening can promote protection against UV damage through enhanced keratinocyte proliferation, independent of an increase in pigmentation [38]. Epidermal density was particularly increased with the combined protocol by 57.7% compared with 7.1% for radiofrequency alone. 

Together, these findings suggest that radiofrequency and HA mesotherapy were able to increase epidermal thickness by enhancing keratinocyte proliferation, with the greatest effects observed when the two procedures were combined. Importantly, no evidence of pathologic hyperkeratosis or an immune response was observed. The exact mechanisms by which these treatments may enhance keratinocyte proliferation have not been confirmed, but research with human skin substitutes subjected to radiofrequency treatment has demonstrated upregulation of keratin genes involved in the epidermal differentiation process with downregulation of matrix metalloprotein genes [39].

Several studies, all of which used crosslinked HA fillers, have reported that radiofrequency treatment prior to HA filler injection may provide synergistic and long-lasting effects for the reduction of nasolabial fold wrinkles [40], marionette lines [41], and forehead lines [42]. One study used a radiofrequency hydro-injector device to simultaneously deliver microneedle intradermal radiofrequency and a non-crosslinked HA to the periorbital area [43]. Improvements in wrinkles, roughness, and pore volume were noted after two to three treatments.

Radiofrequency devices create electrical currents that convert to thermal energy when they encounter tissue resistance and are widely used in the aesthetic field to induce dermal remodeling. As their effects do not depend on the chromophore, but on the electrical properties of the target tissue, they can be safely used in all Fitzpatrick’s skin types. The current study used bipolar radiofrequency, which is designed to treat more superficial levels of the skin than monopolar treatment, and combined it with HA mesotherapy. Mesotherapy is a treatment approach for skin rejuvenation that introduces a range of compounds for optimal biorevitalization. In the current study, the mesotherapy product used combined a highly biocompatible non-crosslinked HA (18 mg/mL) enriched with 0.01% calcium hydroxyapatite (CaHA) and the amino acids glycine and L-proline. Hyperdiluted CaHA is an established biostimulatory agent to improve skin quality and firmness via its ability to induce neocollagenesis [44]. Glycine and L-proline comprise a major portion of the primary amino acid sequence of collagen, and had the greatest effect on collagen production in a comparison of 20 amino acids added to human dermal fibroblasts [45]. The addition of glycine and l-proline to hyaluronic acid has also been shown to improve hydrogel elasticity and osmoregulation in a series of in vitro viscoelastic studies [46]. The synergistic actions of the radiofrequency treatment combined with those of the mesotherapy contents improve epidermal homeostasis [47] and have both visible and preventative skin benefits. These include improved skin appearance (increased firmness, even skin tone, reduced dryness) and improved epidermal barrier function and skin homeostasis to support optimal skin renewal.

The treatment protocol employed radiofrequency immediately prior to HA mesotherapy and not vice versa. This sequence is important as previous research has suggested that radiofrequency treatment after HA filler injection may lead to degradation of the HA, especially if performed on the same day [48]. A systematic review of monopolar and bipolar radiofrequency devices has shown that they produce measurable improvements in skin laxity with an acceptable adverse event profile [49]. The majority of reported complications were minor and transient in nature; major complication rates were higher with the use of monopolar devices than with the use of bipolar devices. In the current study, no adverse events related to treatment were reported.

A major advantage of this study was the split-face design, which allowed objective assessment of the synergistic effect of combination treatment compared with radiofrequency alone. The study was limited by its relatively small sample size. In addition, the duration of follow-up was not long enough to evaluate the long-term effect of the combination treatment. Further studies, with a larger sample size and a long duration of follow-up in subjects with a range of skin types, are now warranted to confirm the synergistic effect and safety of treatment with this novel radiofrequency and HA mesotherapy combination.

## 5. Conclusions

Radiofrequency devices and HA mesotherapy products represent two separate modalities for skin rejuvenation with synergistic effects on epidermal rejuvenation. In this pilot study, in subjects with dry and lax facial skin, a treatment protocol combining radiofrequency with HA mesotherapy was associated with greater improvements in skin hydration, firmness and tone compared with radiofrequency alone. At a cellular level this was linked to increased epidermal thickness and density. Both treatments were well-tolerated and led to improvements in facial appearance. 

## Figures and Tables

**Figure 1 pharmaceuticals-16-01145-f001:**
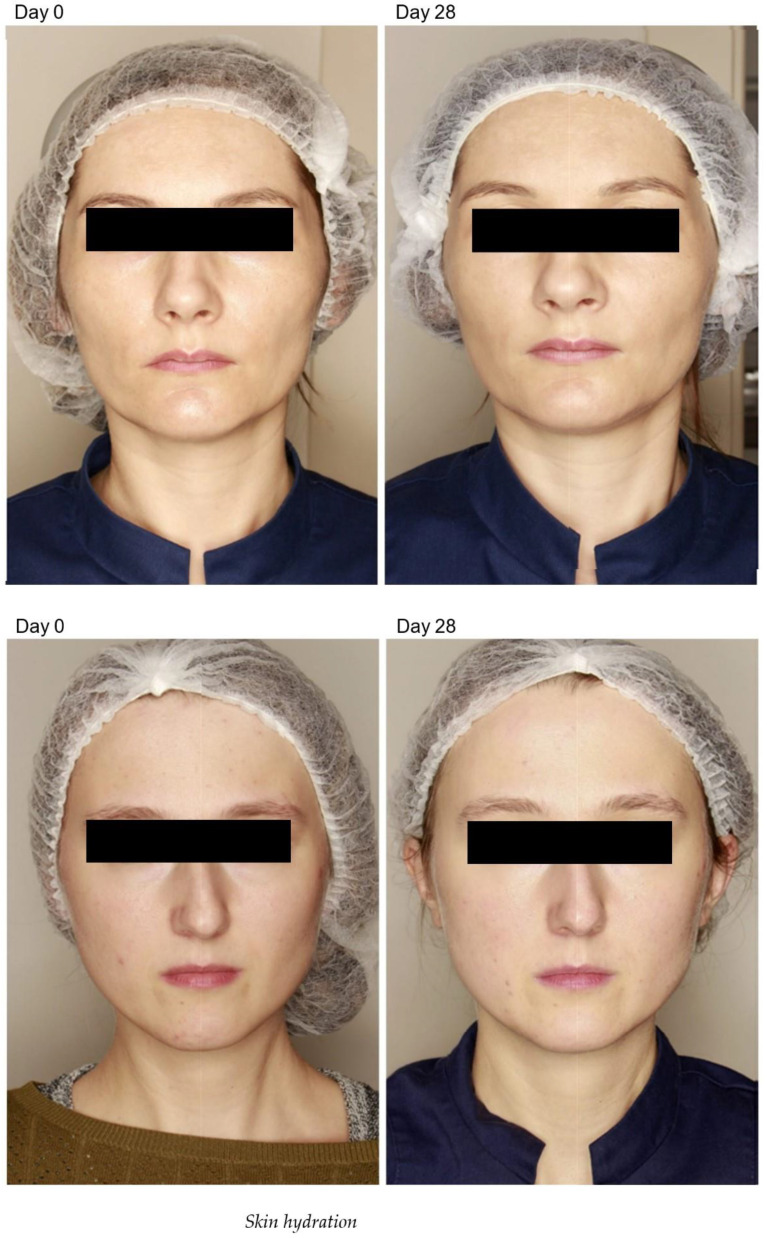
Bipolar radiofrequency and non-crosslinked hyaluronic acid mesotherapy clinical study results. Top image: condition before treatment; condition 28 days after treatment, bottom image: condition before treatment; condition 28 days after treatment.

**Table 1 pharmaceuticals-16-01145-t001:** Changes in parameters of skin barrier function and homeostasis compared with pretreatment (baseline) values.

Parameter	Radiofrequency (% Change)	Radiofrequency + HA Mesotherapy (% Change)
**Cutometer analysis**		
Stratum corneum hydration	+1.0	+23.1
TEWL	+3.9	−5.8
Melanin	−1.5	−7.5
Erythema	+3.0	−7.2
**Smart Mirror-Skin analysis**		
RGB pore	-	−15.6
RGB spots	-	−11.9
UV pigmentation	-	−22.0
UV damage	-	−10.8
Pigmentation changes	-	−27.4
**Ultrasound analysis**		
Change in skin thickness	+1.4	+2.7
Change in epidermal thickness	+5.6	+12.0
Change in skin density	−18.1	+12.4
Change in epidermal density	+7.1	+57.7

## Data Availability

The data presented in this study are available on request from the corresponding author. The data are not publicly available due to privacy restrictions.

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
