# Peer review of "Combined Bipolar Radiofrequency and Non-Crosslinked Hyaluronic Acid Mesotherapy Protocol to Improve Skin Appearance and Epidermal Barrier Function: A Pilot Study"

_pharmaceuticals, 2023, doi:10.3390/ph16081145_

Round 1
Reviewer 1 Report
Dear Authors,
The submitted manuscript compared outcomes of epidermal barrier function between serval intervention groups in a 28-day pilot study of facial skin treatment using HA & radiofrequency combination therapy. In/Exclusion conditions and treatment protocol are well described, results are mostly described concisely, and conclusions are supported by results. Some issues remain:
Introduction:
1. The structure of introduction can be improved for example:
Explanation of skin ageing mechanism, HA, collagen…
Changes in skin elasticity laxity, hydration and overall skin tone with aging
Anti-aging therapy: radiofrequency and hyaluronic acid mesotherapy protocols, etc.
2. From line 33 to 57, paragraphs and sentences can be rearranged to improve readability.
3. Line 65, Giving examples of protein will be highly appreciated, e.g., the markers for keratinocytes in basal layer, spinosum, granulosum, stratum corneum.
4. The aim of this study is well described.
Methods:
1. Line 173, format the sentence.
2. Line 176, delete –
- Results and discussion that can be improved:
1. Ethics information provided, but do the subjects consent to unedited pictures being used. Please edit images in figure 1 to make them deidentified. Please place figure legend under the figure as per conventions.
2. Regarding to the claims made in the manuscript and in Table 1. What is the baseline data for all parameters? The presentation of data needs can be improved. What are absolute values in group of radiotherapy and combination therapy corresponding to % change?
3. Skin Hydration was assessed using a Corneometer C+K. Other mechanistic imaging techniques are available as described in https://doi.org/10.1208/s12248-016-9984-0. This could help in triangulation as well as enhance the manuscript. Given skin hydration and increased epidermal thickness are used as outcomes in this study.
4. In some parts of the manuscript, the authors mentioned epidermal differentiation and l homeostasis. Was this in the scopes of the pilot trial? If so, did the ultrasound suggested any improvement with the combination protocol? The hydration of stratum corneum will significantly change the thickness and density of epidermis.
- Specific hints:
The format of this manuscript needs to be improved:
a. Line 25 – 30
b. Line 173 -175
c. Line 324-326
d. Line 341-342
e. Line 342-344
Are these changes of font style necessary? Please check.
5. Please add new literature on skin - https://doi.org/10.1016/j.addr.2022.114293; https://doi.org/10.3389/fddev.2022.957732
Moderate editing for English language and formatting required. Suggestions provided.
Reviewer 2 Report
The manuscript compares the effects of a bipolar radiofrequency treatment alone and in combination with hyaluronic acid mesotherapy by treating both sides of the face with radiofrequency waves and only one face with mesotherapy. The study was performed on twenty female subjects aged 29 to 54 years with dry and lax facial skin. The effects, estimated 28 days after treatment by cutometer analysis, smart mirror-skin analysis and ultrasound analysis were encouraging and indicate that combining radiofrequency with hyaluronic acid mesotherapy was associated with greater improvements in skin hydration, firmness and tone compared than radiotherapy alone.
Remarks:
The manuscript seems more suitable for Cosmetics than for Pharmaceuticals.
The experimental design is incompletes: there is no comparison of hyaluronic acid mesotherapy alone and hyaluronic acid mesotherapy plus radiofrequency treatment to look. Does radiofrequency treatment improves the effect of mesotherapy? The study demonstrated that hyaluronic acid mesotherapy improves the effect of radiofrequency treatment but not vice versa.
Abstract: The aim of the study is not formulated.
The authors use the term “radiofrequency treatment” interchangeably with “radiotherapy” (Line 27 and Table 1). I recommend to use only the first term as “radiotherapy” associates rather with therapy using ionizing radiation as in cancer treatment.
What was the exact output frequency (or wavelength) of the applicator? Radiofrequency is a rather broad range.
The authors report no adverse effects of the treatment. Were there some side affects like headache, anxiety etc.?
Round 2
Reviewer 1 Report
Dear Authors,
Happy with the changes made. No further suggestions.
Reviewer 2 Report
I accept th explanations and amendments